# Fungal Hyphosphere Microbiomes Are Distinct from Surrounding Substrates and Show Consistent Association Patterns

Nhu H. Nguyen[a]

[a]University of Hawai'i at Mānoa, Honolulu, Hawai'i, USA

**ABSTRACT** Mat-forming fungi are common in forest and grassland soils across the world, where their activity contributes to important soil ecological processes. These fungi maintain dominance through aggressive and abundant hyphae that modify their internal physical and chemical environments and through these modifications select for what appears to be a suite of mycophilic bacteria. Here, the bacteria associated with the fungal mats of *Leucopaxillus gentianeus* and *Leucopaxillus albissimus* from western North America are compared to adjacent nonmat substrates. Within the mats, the bacterial richness and diversity were significantly reduced, and the community composition was significantly different. The bacterial community structure between the two fungal hosts was marginally significant and indicated a shared set of bacterial associates. The genera *Burkholderia*, *Streptomyces*, *Bacillus*, *Paenibacillus*, and *Mycobacterium* were significantly abundant within the fungal mats and represent core members of these hypha-rich environments. Comparison with the literature from fungal mat studies worldwide showed that these genera are common and often significantly found within fungal mats, further reinforcing the concept of a mycophilic bacterial guild. These genera are incorporated into a synthesis discussion in the context of our current understanding of the nature of fungal-bacterial interactions and the potential outcomes of these interactions in soil nutrient cycling, plant productivity, and human health.

**IMPORTANCE** Fungi and bacteria are the most abundant and diverse organisms in soils (perhaps more so than any other habitat on earth), and together these microorganisms contribute to broad soil ecosystem processes. There is a suite of bacteria that appears consistently within the physical space called the hyphosphere, the area of influence surrounding fungal hyphae. How these bacteria are selected for, how they are maintained, and what broader ecological functions they perform are subjects of interest in this relatively new field—the cross-kingdom interactions between fungi and bacteria. Understanding their cooccurrence and their interactions can open new realms of understanding in soil ecological processes with global consequences.

**KEYWORDS** bacterial-fungal interactions, BFI, shiro, fungiphile, microbiome, health, hyphosphere

**Editor** Frédérique Reverchon

Address correspondence to nhu.nguyen@hawaii.edu.

The author declares no conflict of interest.

Forest and grassland soils have an abundance of fungi that are often found as a dense and visible profusion of hyphae and rhizomorphs called fungal, hyphal, or mycelial mats. These mats can be easily distinguished from the surrounding litter or soil (1, 2) and can cover 10 to 20% of the forest topsoil (3). In temperate ectomycorrhizal forests, the fungal mats are composed predominantly of ectomycorrhizal fungi, with a smaller proportion of saprotrophic fungi; there is some indication that fungal mats are also abundant in semitropical forests (3). In grasslands, species that form mats are typically associated with radial zones of outward expanding hyphae, the "fairy rings," often connected to increased plant productivity or plant death (4). The mats

tend to be dominated by fungi in the phylum Basidiomycota, but this growth form is relatively uncommon and polyphyletic across the phylum (2). Mats of fungi in the phylum Ascomycota appear to be widespread (5), but they are often small (several cm in diameter) relative to basidiomycete fungi. Their common and widespread occurrence has led to research into the fungal species that can make these mats, the soil organisms that live in them, the associated biogeochemical transformations, and their broader ecological contributions to soil ecosystems.

The dense concentration of biomass within fungal mats attracts and supports a diversity of soil invertebrates and microbial organisms, including unicellular eukaryotes and bacteria that perform the many biogeochemical transformations of nutrients in forest soils (6, 7). For example, fungal mats can decompose complex organic polymers (2), release bound phosphorous (8), and transform nitrogenous compounds (9). The dense living mycelium is suggested to contribute to maintenance of a living carbon pool belowground, and as these hyphae senesce or are grazed upon by soil fauna and bacteria (6, 10, 11), their necromass may become associated with soil minerals and is fixed within the soil (12, 13). In this sense, the hyphae that make up the mats create a distinct environment that influences the chemical and biological processes around them. Because of their broad distribution and contribution to essential soil processes, fungal mats appear to be an essential component of soil ecosystems where they occur. As such, there has been continuous and increasing interest in the identifying bacterial members that live within fungal mats, how they interact with their fungal hosts, and the outcomes of these interactions.

Characterization of the microbiomes within fungal mats across the world has provided a broad understanding of the community structure and associated biogeochemical transformations. In particular, mats dominated by ectomycorrhizal fungi have been better characterized than their saprotrophic counterparts. For example, the microbiomes of *Piloderma* and *Ramaria* in the Pacific Northwest, USA (14, 15), and those of *Tricholoma matsutake* mat, called "shiro" in Japan and South Korea (16–21), show consistent and comparable patterns. Within these mats, a single fungus dominates the fungal community, and bacterial richness is reduced compared to adjacent nonmat substrates. Similarly, saprotrophic fungi are often just as abundant or more so than mycorrhizal fungi in forest soils (22). Recent work on the microbiomes of saprotrophic mat-forming fungi such as those of *Floccularia luteovirens* (23), *Agaricus arvensis* (24), *Calocybe gambosa* (25), and *Agaricus lilaceps* (26) balances out the literature that had been heavily skewed toward ectomycorrhizal fungal mats. Across all of these studies, bacterial genera *Burkholderia*, *Paenibacillus*, *Mycobacterium*, and *Streptomyces* commonly appeared across the many biomes and fungal hosts. The colonization of mat-forming fungi of a substrate often changes the biogeochemical characteristics of that substrate, such as pH, nutrient availability, and aggregate stability (26), although these changes are less predictable than the typical bacteria inhabiting them.

To further support the concept of a mycophilic bacterial guild within the fungal hyphosphere, this study characterizes bacteria within the mats of two related species, *Leucopaxillus albissimus* and *Leucopaxillus gentianeus*, that occur in forested soils of western North America. These saprotrophic fungal species are members of a genus that is known for producing a profusion of dense mats that can cover an area of over 1 m$^2$ and show similar characteristics to other fungal mats, such as hydrophobicity, but their microbiomes have not been studied. Based on the available literature on bacteria associated with fungal mats, it is predicted that these *Leucopaxillus* mats host bacterial communities that are different from those of surrounding nonmat substrates and host distinct members of the bacterial community commonly seen in fungal mats across the world.

## RESULTS

Both *L. gentianeus* and *L. albissimus* hosts were the dominant fungus in their respective mats (95% and 97%, respectively, of total sequences) (see Fig. S2 to 4 and Table S1

**TABLE 1** Bacterial richness, diversity, and community dissimilarity of mat versus nonmat samples and pairwise comparisons among these samples[a]

| Comparison | ASV richness | Faith's phylogenetic diversity | Community dissimilarity |
|---|---|---|---|
| Mat vs nonmat | $F_{3,20} = 10.1$, $P < 0.001$ | $F_{3,20} = 16.6$, $P < 0.001$ | [b]$F_{1,22} = 2.08$, $P = 0.001$ $R^2 = 0.086$ |
| *L. gentianeus* mat vs *L. gentianeus* nonmat | $P < 0.001$ | $P < 0.001$ | $F_{1,10} = 2.37$, $P = 0.001$ $R^2 = 0.192$ |
| *L. gentianeus* mat vs *L. albissimus* mat | $P = 0.077$ | $P = 0.039$ | $F_{1,10} = 1.52$, $P = 0.011$ $R^2 = 0.132$ |
| *L. gentianeus* mat vs *L. albissimus* non-mat | $P = 0.003$ | $P < 0.001$ | $F_{1,10} = 2.14$, $P = 0.002$ $R^2 = 0.176$ |
| *L. gentianeus* nonmat vs *L. albissimus* mat | $P = 0.074$ | $P = 0.009$ | $F_{1,10} = 1.62$, $P = 0.002$ $R^2 = 0.139$ |
| *L. gentianeus* nonmat vs *L. albissimus* nonmat | $P = 0.715$ | $P = 0.634$ | $F_{1,10} = 1.45$, $P = 0.004$ $R^2 = 0.126$ |
| *L. albissimus* mat vs *L. albissimus* nonmat | $P = 0.435$ | $P = 0.109$ | $F_{1,10} = 1.22$, $P = 0.005$ $R^2 = 0.109$ |

[a]ANOVA and Tukey's HSD test were used to determine differences in ASV richness and Faith's phylogenetic diversity. Unweighted UniFrac distance was used to compare community dissimilarity. *F*-statistics and $R^2$ are reported when available.
[b]Blocked by host species.

in the supplemental material), and they appear to exert a selective force on the bacterial community within (Table 1, Fig. 1 and 2). Mats had significantly lower bacterial amplicon sequence variant (ASV) richness ($P < 0.001$) and phylogenetic diversity ($P < 0.001$) and a community that is different than those within nonmats ($P = 0.001$, $R^2 = 0.086$). However, this was observed only *L. gentianeus* in pairwise comparisons ($P < 0.001$, $R^2 = 0.192$). Bacterial richness and diversity within *Leucopaxillus albissimus* mats were not significantly different from those of nonmats ($P > 0.109$), but the community composition was significantly different ($P = 0.005$, $R^2 = 0.109$). Between the two fungal host species, differences in bacterial ASV richness ($P = 0.077$), phylogenetic diversity ($P = 0.39$), and community composition ($P = 0.011$, $R^2 = 0.132$) were marginally significant.

**Abundant and distinct bacteria within fungal mats.** The mats of each *Leucopaxillus* species contained a core set of taxa with some overlap between the two fungal host species (Tables S2 to S4). The mats of *L. gentianeus* hosted 15 ASVs that can be grouped into *Streptomyces* (1.41%), *Pirellulaceae* taxon Pir4 (0.41%), *Acinetobacter* (0.13%), *Burkholderia* (0.13%), *Cupriavidus* (0.12%), and *Paenibacillus* (0.03%). The mats *L. albissimus* hosted 35 ASVs that can be grouped into *Pirellulaceae* (1.28%), *Pirellulaceae* taxon Pir4 (0.36%), *Lacunisphaera* (0.21%), *Isosphaeraceae* (0.15%), *Nitrosomonadaceae* spp. (0.13%), *Cupriavidus* (0.1%), *Acinetobacter* (0.07%), *Afipia* (0.07%), *Burkholderia* (0.06%), *Paenibacillus* (0.06%), *Achromobacter* (0.04%), and *Acidibacter* (0.03%). Therefore, only *Streptomyces* was unique to *L. gentianeus*, whereas *Isosphaeraceae* spp., *Afipia* spp., *Nitrosomonadaceae* spp., *Acidibacter* spp., and *Lacunisphaera* spp. Were unique to *L. albissimus*. Five other genera represented the core taxa across the two fungal host species (*Pirellulaceae* taxon pir4, *Cupriavidus*, *Acinetobacter*, *Burkholderia*, and *Paenibacillus*), but analysis of composition of microbiomes (ANCOM) only supported *Burkholderia* being significantly present (w-value, 83). It should be noted that ANCOM tends to be conservative, so taxa that do show significance, such as *Burkholderia* here, should be emphasized.

A more sensitive comparison using the DESeq approach showed that some bacterial ASVs had significant differential abundances (log$_2$ fold change) in fungal mats over nonmats (Fig. 3): *Acidobacteriota*, *Bacteroidota* (except one ASV), *Chloroflexi*, Cyanobacteria, *Desulfobacterota*, *Gemmatimonadota*, *Myxococcota*, *Patescibacteria*, and *Verrucomicrobiota* (except one ASV) were found only in nonmat litter. Members of the *Actinomycetota*, *Planctomycetota*, and *Proteobacteria* had differential abundance in both mats and nonmats. Some taxa had high differential abundance in mats and are found exclusively there. Some notable examples of these taxa are members of the *Actinomycetota* (*Streptomyces* [3.09%, 2 ASVs in nonmats] and *Mycobacterium* [0.47%]), members of the *Firmicutes*

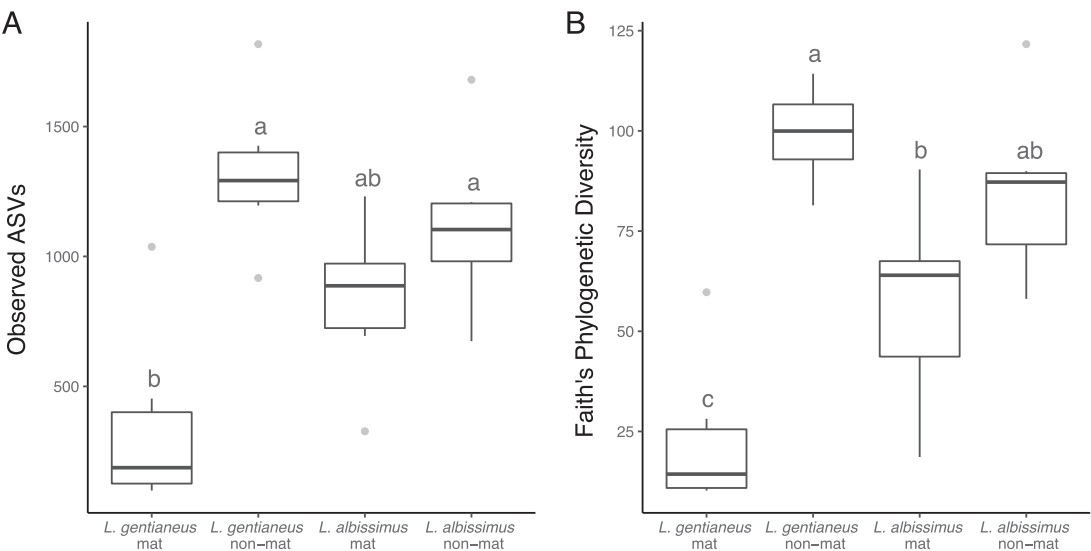

**FIG 1** (A and B) Bacterial (A) observed richness and (B) Faith's phylogenetic diversity compared among mat and nonmat samples across the two fungal host species. Letters above the boxes indicate significance groups.

(*Bacillus* [2.04%] and *Paenibacillus* [0.81%]). members of the *Planctomycetota* (Pir4 [2.79%, 1 ASV in nonmat]). and members of the *Proteobacteria* (*Acinetobacter* [2.35%] and *Burkholderia* [1.27%]). Therefore, members of the genera *Mycobacterium*, *Streptomyces*, *Pirellulaceae* taxon Pir4, *Acinetobacter*, and *Burkholderia* are most consistently present in fungal mats.

## DISCUSSION

Hyphal dominant environments such as fungal mats exert strong influences on the fungal, but especially bacterial, communities within them. The many studies of the shiro, or the mats of *Tricholoma matsutake* (16, 17, 20), *Agaricus gennadii* (27), and *Floccularia luteovirens* (23) in Asia; *Agaricus arvensis* (24), *Calocybe gambosa* (25), and *Laccaria proxima* (28) in Europe; and *Agaricus lilaceps* (26), *Piloderma* spp., and *Ramaria* spp. (14, 15), *Xerocomus pruinatus*, and *Scleroderma citrinum* (29) in North America show consistent and statistically significant patterns: the fungus that formed the mats dominated the fungal community, bacterial richness and diversity decreased, and the bacterial community composition therefore also changed relative to adjacent nonmat

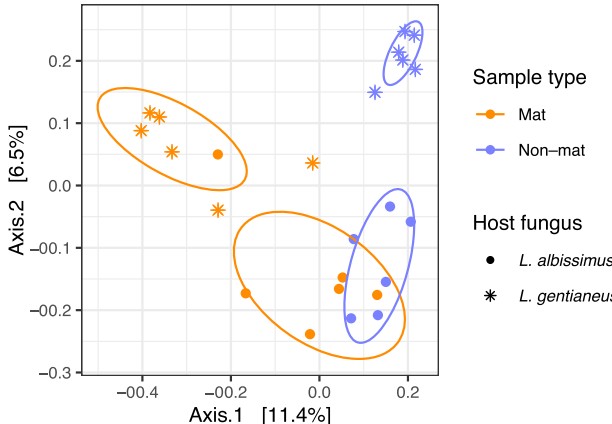

**FIG 2** Principal-coordinate analysis (PCoA) ordination of the bacterial community in mat and nonmat samples across the two different fungal host species. The ellipses are drawn to group sample types and fungal host species. Unweighted UniFrac distance was used for this ordination.

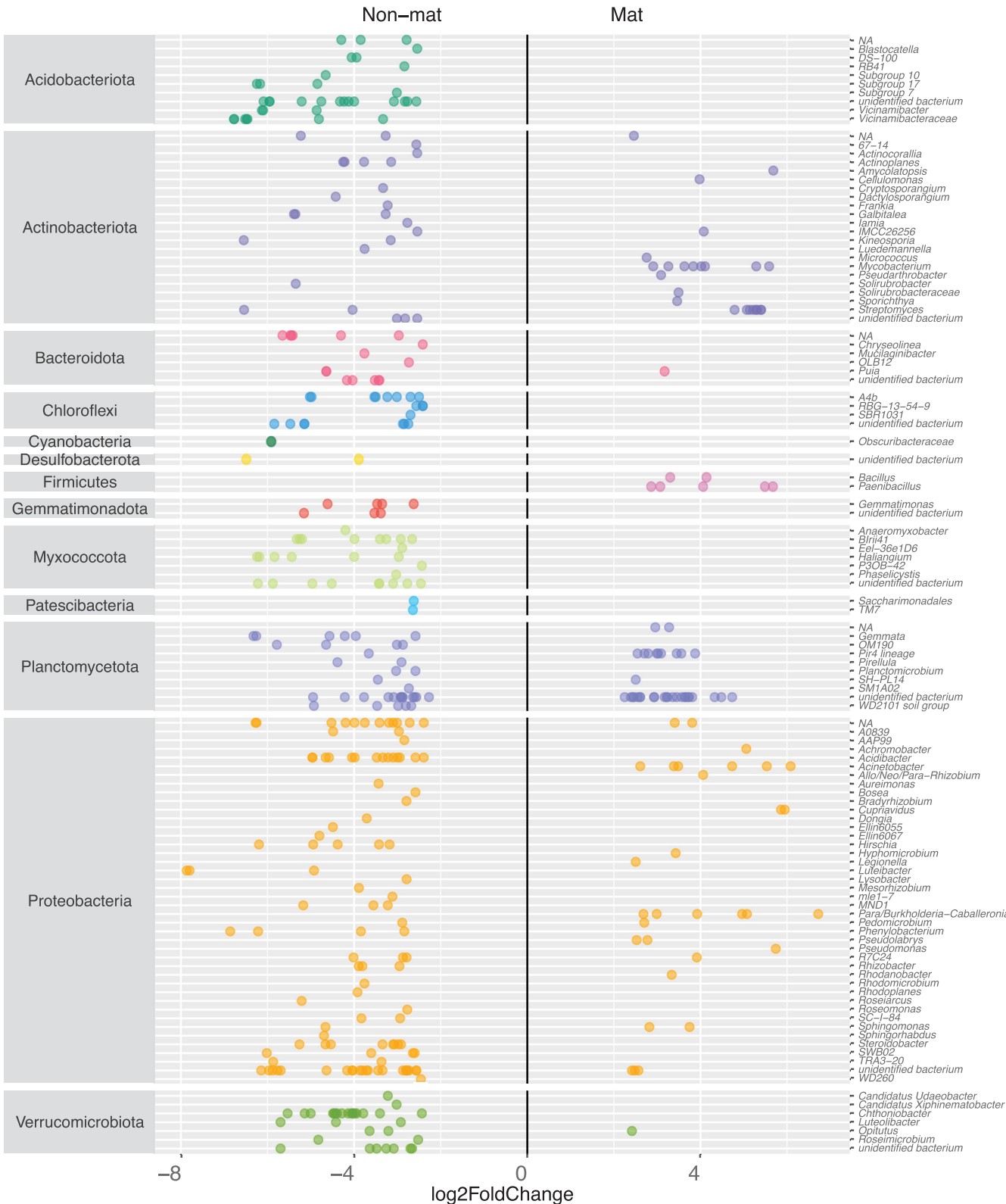

**FIG 3** Significantly abundant bacterial ASVs in nonmat (left) and mat (right) samples for both *L. albissimus* and *L. gentianeus*. Certain genera are found only in mat samples, some taxa are found only in nonmat samples, and some are found in both. Each dot on represents a single ASV, grouped into a genus on each line.

substrates. The results of this study generally align with these patterns from the literature where a single fungus (the host) dominated the fungal community, fungal and bacterial richness and diversity were significantly lowered, and fungal and bacterial community composition significantly shifted compared with adjacent nonmat substrates. Despite the differences in the local environment (e.g., edaphic factors, plant litter chemistry, fungal host species), the consistent outcomes may underlie a widespread selective process(es) on the communities within the fungal mats, irrespective of the trophic guild of the fungi (mycorrhizal or saprotrophic) or whether they occur in forests or grasslands.

**Bacterial genera broadly associated with fungal mats.** The mats of *Leucopaxillus* species host a set of bacteria genera that are indicative of those associated with living hyphae and are consistent with other studies of fungal mats across the three continents mentioned above. Of these genera, *Burkholderia* (*sensu lato*, which includes *Paraburkholderia*) was the most consistent and was found in almost every study reviewed. The other commonly occurring genera were *Streptomyces*, *Bacillus*, *Paenibacillus*, and *Mycobacterium*, but they were not always consistently found across all samples and may reflect the selection based on soil and fungal host species (e.g., habitat filtering) and/or geographic influences (17, 21). For example, several of these core taxa were found only in *L. albissimus*, whereas *Streptomyces* was found only in *L. gentianeus* mats. The patchy abundance of *Streptomyces* may be explained by selection based on host species or actively growing hyphae (30), although a larger and more widespread sampling would be required to fully support this idea. Whether these genera are consistent or specific to a host or environment, their repeated appearance within fungal mats in studies around the world indicate their attraction toward fungal hyphae, and they likely influence the ecology and functional outcomes of fungal hyphae as they traverse the soil environment.

In this study, the medically relevant genera, *Burkholderia/Paraburkholderia*, *Mycobacterium*, and *Acinetobacter* were all significantly present in fungal mats compared to nonmat substrates. The genus *Burkholderia* contains human-pathogenic members, but it appears that *Paraburkholderia* species are environmental and not human-pathogenic (31). Likewise, *Mycobacterium* has been found with fungal mats across various environments (17, 19, 21, 32, 33). The genus contains the human-pathogenic species *M. tuberculosis*, which causes tuberculosis, and *M. leprae*, which causes leprosy. However, environmental strains are often referred to as nontuberculous mycobacteria that cause opportunistic infections, particularly among people with depressed immune systems (34, 35). *Acinetobacter* species are implicated in multidrug resistance and infections in humans, particularly in the hospital environment (36, 37). These bacteria are often reported to be isolated from soil and water as a general substrate source. Their consistent appearance with fungi suggests that, at least in the soil environment, they persist through interactions with fungal hyphae. This provides a clue to the mystery of how these bacteria may persist in the soil and perhaps be dispersed through flowing water and thus could explain how they are often isolated from these substrates.

More specific to this study and atypical of fungal mats worldwide, the Pir4 lineage (*Planctomycetota*; *Pyrellulaceae/Pyrellula*) were significantly abundant in *Leucopaxillus* mats. These unusual bacteria are mostly found in aquatic habitats but have been reported in alpine or arctic calcareous soils (38–40) or host-associated environments (41). Perhaps most relevant here are their associations with lichens (42, 43), part of which are composed of fungi. The phylum *Planctomycetota* was also commonly found as associates of mushrooms (44), but the lineage was not further defined. Their ecology as terrestrial organisms is still relatively unknown, but their occurrence with lichens, fungal mats, and possibly mushrooms suggests a new terrestrial host-associated niche for these unusual bacteria.

It is noteworthy that the bacteria associated with certain dense hyphal environments do not share the same genera found in fungal mats. One of the most well-studied fungal mat systems, those cultivated by fungus-growing ants, shows signatures of animal microbiomes rather than those of mats reported here and elsewhere (45–47).

The patterns of bacteria associated with these cultivated mats are inconsistent, with only *Burkholderia* appearing in many (but not all) of the studies reported. Similarly, the bacteria associated with fungal necromass appear to also be different, with the genera *Burkholderia*, *Chitinophaga*, *Mucilaginibacter*, *Luteibacter*, *Variovorax*, and *Pseudomonas* representing the majority of the genera with the highest relative abundance (10, 33, 48). In living mushroom tissues from a global survey, the genera *Halomonas*, *Serratia*, *Bacillus*, *Cutibacterium*, *Bradyrhizobium*, and *Burkholderia* make up the core microbiome (44). Across all three dense hyphal environments of cultivated, living, and dead hyphae, only the genus *Burkholderia* provided a common link.

**Selection of bacteria within fungal mats.** Mat-forming fungi maintain their dominance and may select for certain members of the bacterial community through physical and chemical modification of their environment. Through physical modification of their hyphae, for example, using hydrophobins, fungal mats maintain low water potentials and decrease nutrient movement across these environments (2, 14). In the *Leucopaxillus* species studied here, the strong hydrophobicity may be a function of both hydrophobins and hyphal ornamentation with calcium oxalates (Fig. S5). Active fungal mats tend to have one lower-pH unit relative to nonmat environments, and although many papers report these decreases to be statistically significant, they were also recognized to be within the physiological buffer range of many mesophilic bacteria and likely not big enough to drive community changes (15, 49, 50).

Analogous to the rhizosphere, the hyphosphere (soils influenced by hyphae) is a nutrient hot spot where fungal hyphae release various labile compounds such as sugars, sugar alcohols, organic acids, and amino acids that attract bacteria (51, 52). Fungal mats can secrete large amounts of oxalic acid, among other organic acids (53), and oxalic acid may be one of the mechanisms that select for the specific sets of bacteria within these mats. *Leucopaxillus* species secrete large amounts of oxalic acid that complexes with calcium to form calcium oxalate outside their hyphae (Fig. S5). Within *Leucopaxillus* mats, *Streptomyces*, *Burkholderia*, *Mycobacterium*, *Bacillus*, *Paenibacillus*, and *Cupriavidus* were significantly abundant, and these bacteria are able to metabolize oxalates and, in some instances, use them as a sole carbon source (30, 54–56). This suggests that fungal exudation and bacterial consumption of oxalates could play a role in selecting for these bacteria within *Leucopaxillus* fungal mats.

**The functional outcomes of interactions in fungal mats.** In environments with an abundance of hyphae, it can be expected that both positive and negative interactions occur between the host fungus and bacteria that can contribute to soil nutrient cycling and promote plant growth. Actively growing fungal mats (such as those reported in forests of the Pacific Northwest and those of fairy rings in grasslands) have increased activity and nutrient availability (8, 25, 26) that are often attributed to fungal decomposition activities, but bacteria can also contribute to this process. For instance, nectrotrophic and biotrophic interactions between fungi and bacteria could liberate nutrients from fungal hyphae (57). Within *Leucopaxillus* mats, several genera, including *Bacillus* and *Paenibacillus*, have been implicated in mycophagy (obtaining nutrients from living hyphae). These genera are common inhabitants of plant rhizospheres in agricultural soils and can act as plant-growth-promoting bacteria (58). The mechanism of plant growth promotion likely varies, but these bacteria have a large repertoire of enzymatic capabilities and functional traits (59–61) and may use them to decompose fungal necromass (33, 61). If these bacteria do indeed antagonize fungi and destroy their cells through hydrolytic enzymes (61–63), they participate in release of nutrients that can then be captured by plants and therefore contribute to plant productivity as plant-growth-promoting bacteria.

Although much of the literature is focused on the antagonistic interactions between fungi and bacteria, positive or neutral interactions have been described, such as the ability of bacteria to use fungal hyphae as "highways" that resulted in overall positive functional outcomes. Recent works showed that rhizobia, in particular, *Mesorhizobium*, take advantage of fungal highways to reach their legume partners and efficiently nodulate them (64, 65). Although present but not significantly abundant in *Leucopaxillus* mats, *Mesorhizobium* species have been found in ectomycorrhizal root tips colonized by fungi

(29, 66). In this study, *Cupriavidus*, a motile member of the *Burkholderiaceae* that is able to nodulate *Mimosa* species (67), were significantly abundant within *Leucopaxillus* mats. They may persist within these mats by metabolizing fungal-derived compounds such as calcium oxalate as discussed above and traverse these hyphae to connect with a plant partner in which to colonize. Other functional contributions from this positive perspective remain to be described (see reference (68) for a comprehensive discussion).

**Burkholderia are persistent and intimate associates of soil fungi.** The pattern of occurrence between *Burkholderia* with basidiomycete fungi is clear and well supported across many fungal mat studies worldwide, but the functional role of these mycophilic *Burkholderia* remains to be explained. However, the numerous comparative studies of *Burkholderia* species provided some clues as to the nature of the association (69, 70). In general, *Burkholderia sensu lato*, have comparatively large genomes to accommodate the genes that allow for flexibility and versatility in their ecologies (70). They are typically host associated, and although there are some groups that are beneficial (such as the plant-nodulating, nitrogen-fixing endophytes, or endohyphal symbionts), many appear to have pathogenic associations with their hosts. In particular, a number of species are able to produce potent antifungals that act against plant- and animal-pathogenic *Beauveria*, *Botrytis*, *Fusarium*, *Metarhizium*, *Rhizoctonia*, *Sclerotinia*, *Verticillium*, and more broadly, *Phytophthora and Pythium* (71–74). Within the fungal mats tended by ants, *Burkholderia* can also inhibit the growth of the parasitic fungus *Escovopsis weberi*, but the host fungus *Leucoagaricus gongylophorus* remains unaffected (74, 75). This was interpreted as a specific interaction within the ant-fungus-parasite system. Evidence thus far points to the consistent occurrence of *Burkholderia* in basidiomycete fungal mats, and their antifungal properties against ascomycete fungi suggests that perhaps there are selective interactions with different taxonomic groups of fungi (although it should be noted that there is a bias toward using plant-pathogenic ascomycetes in antifungal bioassays). In the complex soil environment where diverse groups of fungi exist and competition for resources is fierce, being able to consume the exudates released from one fungal host while decomposing the hyphae of another host can be an advantageous strategy. The genomic signatures and extracellular products of these fungi discovered so far suggest broad modes of interactions that span the symbiotic spectrum, and it is likely that different soil *Burkholderia* species will interact with fungi across this spectrum.

**Conclusions.** This study, in combination with other studies elsewhere, revealed a consistent and clear pattern of the diversity and composition of fungi and bacteria that inhabit fungal mats and more clearly defined the genera of mycophilic bacteria. The colonization of a mat-forming fungus almost always resulted in the reduction of the fungal and bacterial diversity and significantly shifted community composition. Within these fungal mats, *Burkholderia*, *Streptomyces*, *Bacillus*, *Paenibacillus*, and *Mycobacterium* were commonly found, and this reinforces the concept of a mycophilic bacterial guild. *Burkholderia* was the only genus consistently present in fungal mats, along with all other fungal-associated environments such as ant nests, mushrooms, and fungal necromass, suggesting its intimate association with fungi in the soil. Evidence and interpretations from the literature suggest that the interactions occur both ways, with the fungi providing suitable substrates and the bacteria making use of those substrates across the symbiotic spectrum. However, most of the mechanisms of interactions and maintenance of these bacteria within fungal mats discussed here should be considered hypotheses, as they have not been extensively tested and validated. This leaves a broad range of stimulating possibilities to experimentally test the nature of fungal-bacterial interactions in fungal mats, the dynamics of their associations, and the consequences of their interactions to larger ecological processes in human-associated, managed, and natural ecosystems.

## MATERIALS AND METHODS

**Site and sampling.** The samples were collected from two field sites in December 2011. *Leucopaxillus gentianeus* mats (Fig. 4A) were collected within the San Francisco watershed (37.498397° −122.355608°)

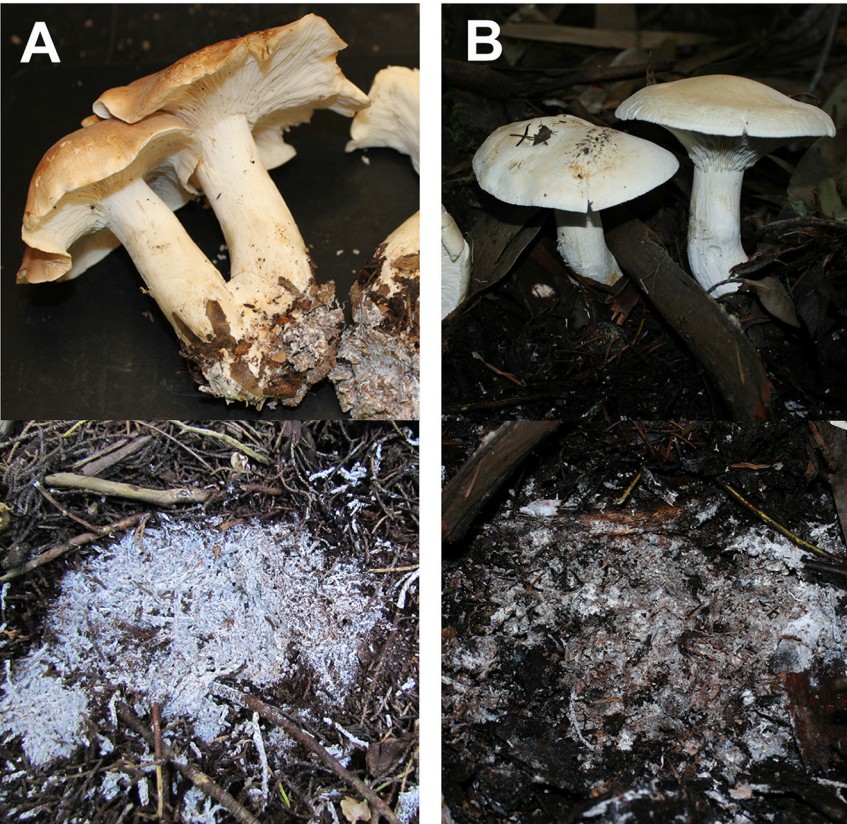

**FIG 4** (A and B) *Leucopaxillus gentianeus* (A) and *Leucopaxillus albissimus* (B) mushrooms form dense hyphal mats beneath the mushrooms in the litter. An edge of a mat was easily distinguished when there was an abrupt ending to mycelial colonization, much like the edge of a fungal colony growing in a petri plate.

under a monodominant stand of Monterey Cypress (*Hesperocyparis macrocarpa*). *Leucopaxillus albissimus* mats (Fig. 4B) were collected at Point Reyes National Seashore (38.043889° −122.804094°) in a stand of Blue Gum Eucalyptus (*Eucalyptus globulus*). The litter substrate is composed of nearly 100% leaves and twigs from the associated trees. At each site, two plots (approximately 30 m apart) were determined where the fungus of interest was fruiting. Immediately underneath the mushrooms, surface litter was gently removed to reveal the dense mycelial mat underneath (Fig. 4). Three mat samples (10 by 5 by 5 cm) within 10 m of each other were sampled. Three nonmat litter samples of the same dimensions and approximately 0.5 m away from the edge of a mat were also sampled. Care was taken so that only leaf litter of similar depths was collected without any of the mineral soil beneath the litter. This was usually not difficult because the litter layer could be as thick as 10 cm. A total of 24 samples (12 mat and 12 nonmat) were collected within 8 days of each other, air-dried, homogenized, and kept frozen until DNA could be extracted.

**Molecular methods.** DNA was extracted from 1 g of each sample by washing with 1 mL of sorbitol wash buffer and further homogenizing them in 200-$\mu$m silica beads for 5 min at maximum speed on a Vortex Genie 2 (Thermo Fisher, Inc., USA). Due to the high concentrations of polyphenols in these samples, the homogenates were washed several times in sorbitol wash buffer (76) prior to suspension in 2% cetyltri-methylammonium bromide (CTAB) buffer with 2% PVP-40. DNA was extracted following standard chloroform precipitation, isopropanol precipitation, and ethanol dehydration and then further cleaned using solid-phase reversible immobilization (SPRI) magnetic beads. The cleaned DNA was stored at −20°C.

The V4 region of the bacterial 16S gene and the internal transcribed spacer 2 (ITS2) region of the fungal ITS rRNA gene were amplified using a dual-PCR, paired-barcode approach (77). In brief, the first PCR amplifies the gene of interest with the primer pair 515F (78) and 806R (79) and the ITS primer pair ITS5.8S-fun and ITS4-fun (80) using the following PCR parameters: polymerase activation at 98°C for 30 s, followed by 29 cycles of 98°C for 10 s, 55°C for bacteria and 53°C for fungi for 15 s, 72°C for 10 s, and a final extension at 72°C for 7 min. Each PCR product was cleaned using SPRI magnetic beads, and 1 $\mu$L of the PCR product was used as a template for the second PCR that attaches the barcode, as well as Illumina i5 and i7 adapters, to each end of the PCR product using the following PCR parameters: polymerase activation at 98°C for 30 s, followed by 14 cycles of 98°C for 10 s, 52°C for 15 s, 72°C for 10 s, and final extension at 72°C for 7 min. High-fidelity and proof-reading polymerase NEBNext Ultra II Q5 master mix (New England Biolabs, USA) was used for all amplification steps. The final products were cleaned using SPRI magnetic beads and quantified using a Qubit 3 fluorometer. All products were combined at

equimolar concentration; negative and positive controls were also included. The final library was sequenced using Illumina MiSeq PE300 chemistry.

**Bioinformatics.** Sequence data processing and quality control were performed using QIIME 2 v2021.11 (81). Raw sequences were demultiplexed and truncated, and primers, adapters, and those with a q value of <30 were removed. They were then paired and denoised using the DADA2 plugin (82). DADA2 amplicon sequence variants (ASVs) were used for further analyses. The sklearn classifier was used to identify ASVs trained on the SILVA v138 database. Sequences classified as "unassigned," "mitochondria," "chloroplast," or "Eukaryota" were removed. Archaeal sequences made up 0.007% of the total sequences and thus were considered background noise and were also removed. The negative PCR control showed low occurrences (0.06%) of taxa that do not occur in the experimental samples and thus were removed. For the fungal data set, ASVs were further clustered into operational taxonomic units (OTUs) at 97% sequence similarity using VSEARCH to best reflect the membership of the mock community (83). The negative-PCR control had a small portion (0.0039%) of two OTUs that appeared in the actual samples, indicating some minor tag bleeding, despite having dual-barcoded samples. However, the number of sequences was minor, and this tag bleeding was not consistent across all samples; therefore, the negative control was simply removed prior to processing. Fungal OTUs were identified using the same classifier approach trained on the UNITE v8.3 database (84). Singletons were removed from both data sets. Bacterial samples were rarefied to 18,640 sequences, and fungal sequences were rarefied to 19,500 sequences (Fig. S1). The core taxa within and across all mat samples were determined by the presence of taxa that occurred in 50% or more of the samples and have sequence abundance greater than 0.01% (85). Contingency tables and distance matrices created in QIIME 2 were exported for further analyses in R.

**Statistical analysis.** Statistical analyses were performed in R v4.2.0 (86). Observed richness and phylogenetic diversity were compared using analysis of variance (ANOVA), followed by the Tukey honestly significant difference (HSD) test. Community clustering was visualized in ordination space using principal-coordinate analysis. Community composition was statistically compared using *adonis2* in the vegan package (87) using Bray-Curtis distances for fungi and unweighted UniFrac distances for bacteria. Analysis of bacteria using Bray-Curtis distances showed similar outcomes. Bacterial ASVs that significantly differed in abundance across the sample types (in this case mat versus nonmat) were measured using the analysis of composition of microbiomes (ANCOM) plug-in for QIIME 2 (88). For a more granular measure of differentially abundant taxa between sample types, the $\log_2$ fold change of ASVs was measured using the DESeq2 package (89).

**Data availability.** The pipeline for QIIME and R analyses, including DNA extraction protocols, is available on the GitHub repository (https://github.com/nnguyenlab/leucopaxillus-mat-microbiome). Sequence data were deposited in the Sequence Read Archive under BioProject number PRJNA893706.

## SUPPLEMENTAL MATERIAL

Supplemental material is available online only.
**SUPPLEMENTAL FILE 1**, PDF file, 1.1 MB.
**SUPPLEMENTAL FILE 2**, XLSX file, 0.01 MB.
**SUPPLEMENTAL FILE 3**, XLSX file, 0.01 MB.
**SUPPLEMENTAL FILE 4**, XLSX file, 0.01 MB.

## ACKNOWLEDGMENTS

I thank my graduate advisor Tom Bruns, who supported unrestricted exploration of this work, and undergraduate student Andrew Lin for building the fungal amplicon libraries.

The National Science Foundation NSF GRFP supported early exploration of this work; implementation was supported by the U.S. Department of Energy, Office of Science, Office of Biological and Environmental Research, Genomic Science Program, under award number DE-SC0020163, and the U.S. National Institute of Health Center of Biomedical Research Excellence project 5P20GM125508-03.

I have no relevant financial or nonfinancial interests to disclose.

N.H.N. performed the field work, prepared the samples, analyzed the data, and wrote the manuscript.

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
