## [Reviewer comments · Microbiology Spectrum]

Microbiology Spectrum

Fungal hyphosphere microbiomes are distinct from surrounding substrates and show consistent association patterns

Nhu Nguyen

Corresponding Author(s): Nhu Nguyen, University of Hawai'i at Mānoa

Review Timeline:

Submission Date:	November 17, 2022
Editorial Decision:	February 13, 2023
Revision Received:	February 21, 2023
Accepted:	February 22, 2023

Editor: Frédérique Reverchon

Reviewer(s): Disclosure of reviewer identity is with reference to reviewer comments included in decision letter(s). The following individuals involved in review of your submission have agreed to reveal their identity: camille truong (Reviewer #1); Adriana L Romero-Olivares (Reviewer #2)

Transaction Report:

DOI: <https://doi.org/10.1128/spectrum.04708-22>

February 13, 2023

Dr. Nhu H Nguyen
University of Hawai'i at Mānoa
Tropical Plant and Soil Sciences
3190 Maile Way
St. John 102
Honolulu, HI 96822

Re: Spectrum04708-22 (Fungal hyphosphere microbiomes are distinct from surrounding substrates and show consistent association patterns)

Dear Dr. Nhu H Nguyen:

Two independent reviewers have now provided their comments on your manuscript. Their assessment was quite contrasting, as you will see. I am prepared to evaluate a revised manuscript as I think this is an interesting topic. However, concerns expressed by Reviewer #1 are very relevant and should be thoroughly addressed: exploratory results are valid as long as they are presented as such, and conclusions should be limited to what is shown by the results.

Link Not Available

Sincerely,

Frédérique Reverchon

Journals Department
Reviewer comments:

Reviewer #1 (Comments for the Author):

This work aligns with previous studies showing that fungal mats harbour distinct bacterial communities and reduced diversity compared to adjacent litter. The analyses are well-performed, but the sampling size is rather small to draw general conclusions

about the widespread occurrence of specific bacteria in fungal mats. The results are for the most part descriptive, and the discussion mostly provide suggestions that are not based on evidence tested here.

For example, the results don't clearly demonstrate that the two focal species harbour similar patterns of bacterial diversity and community structure, despite growing in different habitats (Fig. 2 and 3). As the author stated in the conclusion, only Burkholderia was consistently present in fungal mats and the association of fungal mats with other bacterial genera is not clearly demonstrated.

The author also emphasises how widespread and abundant fungal mats are but mostly mentions studies from temperate forests of the northern hemisphere, especially North America. What about in other regions?

Additional comments are below:

Line 32.

Do all fungi in the hyphosphere produce fungal mats? Please clarify

Line 37.

I suggest mentioning what "shiro" is in the summary.

Lines 42-47.

A more detailed definition of fungal mats, based on current knowledge, would be helpful for the reader and to place the interpretation of results in a more general context. For example, do most soil fungi form fungal mats or is this restricted to a few species, genera or families? The author only mentions basidiomycete mushroom species, what about in ascomycete and other phyla? All the studies mentioned are from North America (or northern temperate regions below). What about other regions of the world?

Line 87.

What is the rationale for selecting two species in different habitats given the hypotheses to be tested?

Line 102

Eucalyptus globulus is an exotic species in California. This should be taken into account in the interpretation of results.

Line 106

If the three mat samples were collected within 10 m of each other in each plot, they may potentially come from the same individual/genet or at least the same population. Do you expect a greater variability from different populations? Please discuss further.

Line 109

What about soil from the organic horizon?

Line 201

What about differences in bacterial mat community structure between the two species?

Line 238

What about habitat filtering? Do you expect to find more similar bacterial communities if the two species grew in the same habitat? What about the same species in different forest types?

Line 307

Do all mat-forming fungi produce oxalate exudation? If this is specific to Leucopaxillus, this would not align with the statement that similar bacterial communities are found in fungal mat worldwide.

Line 331

This is interesting but non relevant here since Mesorhizobium was not a major component of Leucopaxillus mats.

Reviewer #2 (Comments for the Author):

This is a very interesting study investigating the microbiome of fungal mats and how they compare to non-mats. The author found that mats and non-mats vary in their bacterial diversity and proposed the idea of mycophilic bacteria and guilds that grow in close connection with fungal mats. This idea is supported with data that show specific bacteria that inhabit mats and which were not found in non-mats. I really enjoyed reading this explorative paper as it opens up multiple lines of research to further explore, such as questions about metabolomics of fungal-bacterial interactions in fungal mats.

I have a few comments:

23 - Small typo, missing word "discussion in the context our current understanding"

170-171 - could the author very briefly explain what does this analysis does/shows?

177 - Supplementary figure 3 is a more compelling figure that showing the differences in the community of mat vs non-mat than figure 2. Without looking at supplementary figure 3, which many reader will probably not see, it is difficult to get some context for figure 2. Because of this, I suggest the author moves supplementary figure 3 to the main text.

251-252 - typo, "The genus and contains..."

256-257 - I think this is very interesting. Another hypothesis could be that when growing in relation with fungi and other microbes, their virulence factors aren't being expressed; but when growing in isolation they do. Fungal selection forces/competition may be playing an interesting role in regulating virulence.

369 - Should it say mycophilic bacteria?

Staff Comments:

Preparing Revision Guidelines

Please return the manuscript within 60 days; if you cannot complete the modification within this time period, please contact me. If you do not wish to modify the manuscript and prefer to submit it to another journal, please notify me of your decision immediately so that the manuscript may be formally withdrawn from consideration by Microbiology Spectrum.

I thank the two reviewers for their comments that helped to distinguish more clearly what was in this study vs. what was from the literature. The revised manuscript should now clearly state what was from this study and when a comparison was made between this study and the broader literature.

Reviewer comments:

Reviewer #1 (Comments for the Author):

This work aligns with previous studies showing that fungal mats harbour distinct bacterial communities and reduced diversity compared to adjacent litter. The analyses are well-performed, but the sampling size is rather small to draw general conclusions about the widespread occurrence of specific bacteria in fungal mats. The results are for the most part descriptive, and the discussion mostly provide suggestions that are not based on evidence tested here.

I agree that the results are for the most part descriptive and I have been very careful throughout the manuscript to state it as such (and have made some edits based on this review to be even more clear on the approaches).

- For instance, in the last paragraph of the Introduction (Lines 89-91), I stated that “...*this study characterizes bacteria within the mats of two related species Leucopaxillus albissimus and Leucopaxillus gentianeus that occur in forested soils of Western North America*”.
- In the same paragraph (Lines 94-97) I also stated that “*Based on available literature on bacteria associated with fungal mats, it is predicted that these Leucopaxillus mats host bacterial communities that are different from surrounding non-mat substrates and host distinct members of the bacterial community commonly seen in fungal mats across the world.*”
- In the Conclusion (Lines 386-390) I stated that “*Evidence and interpretations from the literature suggests that the interactions occur both ways, with the fungi providing suitable substrates and the bacteria making use of those substrates across the symbiotic spectrum. However, most of the mechanisms of interactions and maintenance of these bacteria within fungal mats discussed should be considered hypotheses as they have not been extensively tested and validated.*”

I agree that the sampling size of the two *Leucopaxillus* species alone can very much limit what can be said. However, I compared the results from the *Leucopaxillus* data with 12 other studies across Europe, Asia, and North America and found similar taxa that occurred in fungal mats and not in non-mat substrates. I also compared these results with 10 other studies in other hyphal-dense environments such as gardens of fungus-growing ants, fungal necromass, and mushrooms. In total, at least 22 studies were used for direct comparison which I feel is an acceptable number of studies, given that this is a relatively new field that is not yet repleated with relevant literature.

I also agree that much of the discussion was based on suggestions gained from reviewing the literature and not tested here, but that was the approach I chose to write this paper (hybrid between primary research and literature review) because it is needed for this new area of study into fungal-bacterial interactions. In the conclusion, I stated that “*Evidence and interpretations*

from the literature suggests that the interactions occur both ways, with the fungi providing suitable substrates and the bacteria making use of those substrates across the symbiotic spectrum. However, most of the mechanisms of interactions and maintenance of these bacteria within fungal mats discussed should be considered hypotheses as they have not been extensively tested and validated.” to recognize that these are patterns and they remained to be validated through experimental approaches.

For example, the results don't clearly demonstrate that the two focal [host] species harbour similar patterns of bacterial diversity and community structure, despite growing in different habitats (Fig. 2 and 3). As the author stated in the conclusion, only *Burkholderia* was consistently present in fungal mats and the association of fungal mats with other bacterial genera is not clearly demonstrated.

The current edits to the manuscript should now adequately show that the two species share some similar patterns:

Lines 187-189: “Between the two fungal host species, bacterial ASV richness ($p = 0.077$), phylogenetic diversity ($p = 0.39$), and community composition were marginally significant ($p = 0.011$, $R^2 = 0.132$).”

Lines 202-204: “Five other genera represented the core taxa across the two fungal host species (*Pirellulaceae* taxon *pir4*, *Cupriavidus*, *Acinetobacter*, *Burkholderia* and *Paenibacillus*), but ANCOM analysis only supported *Burkholderia* being significantly present (w -value 83).”

It should be pointed out that ANCOM, much like indicator species analysis, must correct for multiple comparisons across large microbiome datasets so very few taxa can be detected as significant. Therefore, I have tried to stress this point with the addition of the following sentence in Lines 205-206: “It should be noted that ANCOM analysis tend to be conservative so taxa that do show significance, such as *Burkholderia* here, should be emphasized.”

The author also emphasises how widespread and abundant fungal mats are but mostly mentions studies from temperate forests of the northern hemisphere, especially North America. What about in other regions?

Despite my efforts scour the literature for fungal mat studies from the southern hemisphere, I was not able to find any. The only study that is related was from a broad sampling of mushroom-associated bacteria, which was cited [reference 58]. The other studies came from Europe and Asia.

Additional comments are below:

Line 32.

Do all fungi in the hyphosphere produce fungal mats? Please clarify

The sentence had the wrong subject that caused confusion. “Mats” should not appear here. It has been corrected to: “How these bacteria are selected for, how they are maintained, and what broader ecological functions they perform are subjects of interest in this relatively new field”

To answer the reviewer’s question, the hyphosphere is the area surrounding any fungal hyphae, defined in line 32. All fungal mats are made from hyphae, so all mats contain a hyphosphere.

Line 37.

I suggest mentioning what "shiro" is in the summary.

Since this is a more specific term to eastern Asia, I would like to define it better in line 74 and keep it in the introduction instead of the summary.

Lines 42-47.

A more detailed definition of fungal mats, based on current knowledge, would be helpful for the reader and to place the interpretation of results in a more general context. For example, do most soil fungi form fungal mats or is this restricted to a few species, genera or families? The author only mentions basidiomycete mushroom species, what about in ascomycete and other phyla? All the studies mentioned are from North America (or northern temperate regions below). What about other regions of the world?

Re: Definition

Lines 42-43 defines what is a fungal mat, lines 43-55 defines the characteristics of these mats.

Re: Taxonomic affiliation

This is a good suggestion. The following sentence has been added to lines 49-52: "*The mats tend to be dominated by fungi in the phylum Basidiomycota, but this growth form is relatively uncommon and polyphyletic across the phylum (2). Mats of fungi in the phylum Ascomycota appears to be widespread (5), but they are often small (several cm in diameter) relative to basidiomycete fungi.*"

Re: Breadth of studies

I have not been able to find fungal mat studies in the southern hemisphere, although I am confident that they exist because certain mat-forming ectomycorrhiza taxa also occur in southern beech (*Nothofagus*) forests. Other more widespread mat-forming saprotrophic taxa such as *Chlorophyllum* and *Lepiota* also occur in the southern hemisphere and they also make oxalates (unpublished data).

Line 87.

What is the rationale for selecting two species in different habitats given the hypotheses to be tested?

In this case I had no choice since these two species do not grow in the same habitat. There is only one mat-forming species to be sampled in each habitat. As such, it was not possible to parse out host vs. local environment (in this case plant litter substrate). Based on the evidence, there does seem to be some overlap in community attributes despite this difference (see response above in the general comments). The question of host selection will need to be determined in more controlled experiments.

Line 102

Eucalyptus globulus is an exotic species in California. This should be taken into account in the interpretation of results.

I have modified the sentence on lines 235-238 to try to address the concerns above and here about host variability. "*Despite the differences in the local environment (e.g. edaphic factors,*

plant litter chemistry, fungal host species), the consistent outcomes may underlie a widespread selective process(es) on the communities within the fungal mats, irrespective of the trophic guild of the fungi (mycorrhizal or saprotrophic), or whether they occur in forests or grasslands.”

Line 106

If the three mat samples were collected within 10 m of each other in each plot, they may potentially come from the same individual/genet or at least the same population. Do you expect a greater variability from different populations? Please discuss further.

This is an interesting point, but host population control over their microbiomes is beyond the scope of this study. However, a study referenced [21] in the manuscript showed that the bacterial communities of “shiro” mat of *Tricholoma matsutake* were different from each other based on geographic and soil variables. I would expect that within the same soil environment, there would be little community variability but in a different soil context, the community would be represented by different microbes.

Line 109

What about soil from the organic horizon?

Most of the mat occurred in the organic horizon (leaf litter in our case). I took care not to take samples from the mineral horizon because that would introduce a different environment and introduce unwanted variability into the data (see previous point of response).

Line 201

What about differences in bacterial mat community structure between the two species?

The following sentence has been added to lines 187-189: “*Between the two fungal host species, bacterial ASV richness ($p = 0.077$), phylogenetic diversity ($p = 0.39$), and community composition were marginally significant ($p = 0.011$, $R^2 = 0.132$).*”

Line 238

What about habitat filtering? Do you expect to find more similar bacterial communities if the two species grew in the same habitat? What about the same species in different forest types?

Good point. I have modified the sentence to try and capture the habitat filtering concept in Lines 244-247. “*The other commonly occurring genera were *Streptomyces*, *Bacillus*, *Paenibacillus*, and *Mycobacterium*, but they were not always consistently found across all samples and may reflect the selection based on soil and fungal host species (e.g. habitat filtering) and/or geographic influences (17, 21).*”

Line 307

Do all mat-forming fungi produce oxalate exudation? If this is specific to *Leucopaxillus*, this would not align with the statement that similar bacterial communities are found in fungal mat worldwide.

Likely so. Most of the studies that report on oxalates in fungal mats examined bulk samples that contains a mix of fungal species. In the majority of the mats that I have examined, calcium oxalate can be observed (unpublished data). To address the reviewer’s concerns, which I do agree with, I have modified the sentence (Lines 316-317) to be more specific to *Leucopaxillus* mats: “*This suggests that fungal exudation and bacterial consumption of oxalates could play a role in selecting for these bacteria within *Leucopaxillus* fungal mats.*”

Line 331

This is interesting but non relevant here since *Mesorhizobium* was not a major component of *Leucopaxillus* mats.

Mesorhizobium was needed here as an example of the functional aspect of fungal highways (legume nodulation) and to make connections to another legume nodulator, *Cupriavidus*, that was significantly abundant in our dataset. Therefore, I think it is important to keep it in this paragraph for both content and flow.

Reviewer #2 (Comments for the Author):

This is a very interesting study investigating the microbiome of fungal mats and how they compare to non-mats. The author found that mats and non-mats vary in their bacterial diversity and proposed the idea of mycophilic bacteria and guilds that grow in close connection with fungal mats. This idea is supported with data that show specific bacteria that inhabit mats and which were not found in non-mats. I really enjoyed reading this explorative paper as it opens up multiple lines of research to further explore, such as questions about metabolomics of fungal-bacterial interactions in fungal mats.

Thank you for the nice comments. This paper is meant to open up ideas and areas of research in this field, so I am glad to hear that it stimulated your thinking.

I have a few comments:

23 - Small typo, missing word "discussion in the context our current understanding"
Corrected.

170-171 - could the author very briefly explain what does this analysis does/shows?

I have rephrased the sentence (Lines 172-174) to hopefully it will better explain what the analysis does: "*Bacterial ASVs that significantly differ in abundance across the sample types (in this case mat vs. non-mat) were measured using Analysis of Composition of Microbiomes (ANCOM) plug-in for QIIME 2 (39)*". For more technical information, I would refer the readers to the paper for this method, which explains everything in more details.

177 - Supplementary figure 3 is a more compelling figure that showing the differences in the community of mat vs non-mat than figure 2. Without looking at supplementary figure 3, which many reader will probably not see, it is difficult to get some context for figure 2. Because of this, I suggest the author moves supplementary figure 3 to the main text.

Supplementary Figure 3 shows *fungal* communities rather than bacteria. Since fungal communities is not the core concept of this manuscript, I would like to keep it as a Supplementary Figure.

251-252 - typo, "The genus and contains..."

Corrected.

256-257 - I think this is very interesting. Another hypothesis could be that when growing in relation with fungi and other microbes, their virulence factors aren't being expressed; but when

growing in isolation they do. Fungal selection forces/competition may be playing an interesting role in regulating virulence.

This is an interesting hypothesis into the actual type of interaction with fungi. I think future work in my research group and other research groups will help answer this hypothesis.

369 - Should it say mycophilic bacteria?

Yes, it should. The error has been corrected.

February 22, 2023

Dr. Nhu H Nguyen
University of Hawai'i at Mānoa
Tropical Plant and Soil Sciences
3190 Maile Way
St. John 102
Honolulu, HI 96822

Re: Spectrum04708-22R1 (Fungal hyphosphere microbiomes are distinct from surrounding substrates and show consistent association patterns)

Dear Dr. Nhu H Nguyen:

I am please to tell you that your revised manuscript has been accepted in Microbiology Spectrum. Thank you for taking into account the reviewers' suggestions to improve your article. I am forwarding it to the ASM Journals Department for publication. You will be notified when your proofs are ready to be viewed.

Sincerely,

Frédérique Reverchon
Editor, Microbiology Spectrum
